# Unconventional Functions of Amino Acid Transporters: Role in Macropinocytosis (SLC38A5/SLC38A3) and Diet-Induced Obesity/Metabolic Syndrome (SLC6A19/SLC6A14/SLC6A6)

**DOI:** 10.3390/biom12020235

**Published:** 2022-01-31

**Authors:** Yangzom D. Bhutia, Marilyn Mathew, Sathish Sivaprakasam, Sabarish Ramachandran, Vadivel Ganapathy

**Affiliations:** Department of Cell Biology and Biochemistry, Texas Tech University Health Sciences Center, Lubbock, TX 79430, USA; yangzom.d.bhutia@ttuhsc.edu (Y.D.B.); marilyn.mathew@ttuhsc.edu (M.M.); sathish.sivaprakasam@ttuhsc.edu (S.S.); s.ramachandran@ttuhsc.edu (S.R.)

**Keywords:** macropinocytosis, amino acid-dependent Na^+^/H^+^ exchange, SLC38A5/SLC38A3, enteroendocrine cell, gut hormones, insulin signaling, appetite control, adipocyte differentiation, SLC6A19/SLC6A14/SLC6A6

## Abstract

Amino acid transporters are expressed in mammalian cells not only in the plasma membrane but also in intracellular membranes. The conventional function of these transporters is to transfer their amino acid substrates across the lipid bilayer; the direction of the transfer is dictated by the combined gradients for the amino acid substrates and the co-transported ions (Na^+^, H^+^, K^+^ or Cl^−^) across the membrane. In cases of electrogenic transporters, the membrane potential also contributes to the direction of the amino acid transfer. In addition to this expected traditional function, several unconventional functions are known for some of these amino acid transporters. This includes their role in intracellular signaling, regulation of acid–base balance, and entry of viruses into cells. Such functions expand the biological roles of these transporters beyond the logical amino acid homeostasis. In recent years, two additional unconventional biochemical/metabolic processes regulated by certain amino acid transporters have come to be recognized: macropinocytosis and obesity. This adds to the repertoire of biological processes that are controlled and regulated by amino acid transporters in health and disease. In the present review, we highlight the unusual involvement of selective amino acid transporters in macropinocytosis (SLC38A5/SLC38A3) and diet-induced obesity/metabolic syndrome (SLC6A19/SLC6A14/SLC6A6).

## 1. Introduction

The conventional function of amino acid transporters in mammalian cells is in the maintenance of amino acid homeostasis. Every cell has a need for amino acids from extracellular sources, particularly for the essential amino acids that mammalian cells are not able to synthesize. This need cannot be met without the participation of specific transporters in the plasma membrane because of the hydrophilic nature of these amino acids, a feature that prevents their simple diffusion across the hydrophobic lipid bilayer. The transport process mediated by these amino acid transporters is either uniport (i.e., amino acid transfer in one specific direction) or obligatory exchange (i.e., transfer of one amino acid substrate in one direction which is obligatorily coupled to transfer of another amino acid substrate in the opposite direction). Some are simple facilitative transporters with no involvement of any co-transported ion, whereas others are coupled transporters with involvement of one or more ions (Na^+^, H^+^, K^+^ or Cl^−^). Depending on the stoichiometry of the amino acid substrates and co-transported ions, the transport process could be electroneutral or electrogenic. Therefore, the direction of the amino acid transfer mediated by a given amino acid transporter is dictated by multiple factors: concentration gradients for amino acid substrates and co-transported ions as well as membrane potential. These amino acid transporters are not expressed exclusively in the plasma membrane [1,2,3,4]; they are also found in intracellular membranes, particularly in the lysosomal [5,6] and mitochondrial membranes [7]. This makes sense because amino acid needs of cells are met not only by uptake from extracellular sources but also by transfer from lysosomes following proteolysis in conjunction with autophagy, pinocytosis and macropinocytosis. In mitochondria, many of the biochemical pathways that take place in the matrix involve amino acids (e.g., urea cycle, malate–aspartate shuttle), which requires the transporter-mediated transfer of amino acids in both directions across the inner-mitochondrial membrane.

There are numerous amino acid transporters in mammalian cells, and they differ in substrate selectivity, transport mechanism, driving forces and tissue expression pattern. In terms of the Human Genome Nomenclature, these transporters belong to ten different SLC (solute carrier) gene families (1, 6, 7, 16, 17, 25, 36, 38, 43 and 66) along with four non-transporter proteins (SLC3A1/rBAT, SLC3A2/4F2hc/CD98, ACE2 and collectrin) that serve as chaperones for some of these transporters [1,2]. Depending on their cell-type specific expression, they function not only in the cellular uptake of amino acids but also in transcellular transfer of amino acids across the barrier structures such as the blood–brain barrier, blood–retinal barrier, maternal–fetal barrier and in the absorption of dietary amino acids (intestine) and re-absorption-filtered amino acids (kidney). Loss-of-function mutations in many of these transporters cause specific genetic diseases (e.g., Hartnup disease, cystinuria) [8,9,10,11].

## 2. Unconventional Functions of Amino Acid Transporters

Amino acids represent an important class of nutrients in cellular metabolism, serving as building blocks for protein synthesis and functioning in multiple metabolic pathways such as the urea cycle, heme biosynthesis, one-carbon metabolism, glutaminolysis and neurotransmission, to name just a few. Therefore, the cellular need for amino acids is functionally coupled to the expression levels of specific amino acid transporters in the plasma membrane to coordinate the two events, namely the availability and the utilization of amino acids. This coupling involves specific signaling pathways with direct participation of the involved transporter proteins in the functional crosstalk. This led to the coinage of the term “Transceptor” to highlight the non-traditional role of a protein functioning both as a transporter and as a receptor [12]. In mammalian cells, this novel aspect of an amino acid transporter was first noticed for the classical “system A” [13], a Na^+^-coupled transport system for short-chain amino acids and glutamine belonging to the SLC38 family [14]. System A consists of three specific transporters, SLC38A1 [15,16], SLC38A2 [17] and SLC38A4 [18]; among these three, the novel “transceptor” feature has been ascribed only to SLC38A2 [12,13,19,20]. This feature is responsible for the increase in the plasma membrane density of the transporter protein when cells are deficient in amino acids and conversely for the decrease in transporter density in the plasma membrane when cells are sated with amino acids. More recently, another member of the SLC38 family, namely SLC38A9, expressed in the lysosomal membrane, has been shown to link amino acids in the lysosomes and mTORC1 activity [21,22]. A similar feature has also been found for certain members of the SLC36 family of amino acid transporters [23,24]. In all these cases, amino acid transporters not only mediate amino acid transport but also function as amino acid sensors, thus coupling amino acid status within the cells to amino acid transporter density in the plasma membrane to modulate amino acid entry into cells and amino acid-dependent metabolic pathways. This ensures control of amino acid entry into cells in a manner that is appropriate for the amino acid status (deficient or excess) within the cells and also modulation of metabolic pathways that utilize amino acids such that the rates of these pathways are in tune with the magnitude of amino acid delivery into cells.

Another notable non-traditional function of amino acid transporters is their involvement as the cell surface receptors in retroviruses. Interestingly, this function is independent of the role of these transporters in amino acid transfer. This is in contrast to their “transceptor” function where amino acid transport is coupled to cellular signaling. Retroviruses hijack specific amino acid transporters to gain entry into their target cells [25]. In this process, the viruses bind to the external surface of specific transporters, and then the bound complex undergoes endocytosis, consequently delivering the viruses into the cells. Since the amino acid transporters are expressed in a cell type-specific manner, this also provides the molecular basis for target cell selectivity for these viruses. Furthermore, the variable amino acid sequences of a given amino acid transporter among different species dictates the specificity of virus interaction for the species-specific tropism of a given virus.

To date, three amino acid transporters and one transporter chaperone have been shown to serve as cell surface receptors for specific retroviruses (Table 1). The transporters are murine cationic amino acid transporter 1 (Slc7a1) for ecotropic murine leukemia virus (E-MLV) and bovine leukemia virus (BLV) [26,27], human alanine–serine–cysteine transporter 1 (ASCT1 or SLC1A4) for the feline infectious endogenous retrovirus RD-114, baboon endogenous retrovirus (BaEV), and human endogenous retrovirus HERV-W [28] and Alanine-Serine-Cysteine Transporter 2 (ASCT2 or SLC1A5) for baboon endogenous retrovirus (BaEV) and human endogenous retrovirus HERV-W [29,30]. Even though the transport function has nothing to do with the binding of the virus to the transporter protein, it is likely that the interaction impacts the transport function. Since the virus entry via the transporter involves endocytosis, it is possible that the binding of the virus to the transporter results in decreased density of the transporter protein in the plasma membrane, thus negatively affecting the transport function. This is also true with ACE2, the chaperone for the intestinal amino acid transporters SLC6A19 and SLC6A20 [31]; this chaperone protein serves as the primary receptor for the COVID-19 virus SARS-CoV-2 [32,33]. Deletion of Ace2 in mice led to a drastic decrease in the density of the two transporters in the apical membrane of the epithelial cells of the small intestine [34]. Therefore, it is probable that binding of SARS-CoV-2 to ACE2 in the intestine decreases the trafficking of SLC6A19 and SLC6A20 to the apical membrane, consequently decreasing the absorption of amino acids in the small intestine.

There are several other members of the SLC family that are utilized as cell surface receptors for retroviruses [25], but these are not amino acid transporters. This includes the facilitative glucose transporter GLUT1 (SLC2A1), Na^+^/H^+^ exchanger NHE1 (SLC9A1), two phosphate transporters (SLC20A1 and SLC20A2), two vitamin transporters (SLC19A1 and SLC19A2) and the heme transporter SLC49A1.

More recently, two new unconventional functions of amino acid transporters have been brought to light. This includes potentiation of macropinocytosis and regulation of diet-induced obesity (potentiation or protection depending on the transporter involved). The present review focuses on these two most recently discovered non-traditional features of amino acid transporters.

## 3. Macropinocytosis and SLC38A5/SLC38A3

Macropinocytosis is a mechanism for a non-specific fluid-phase uptake in cells, which is distinct from other similar processes such as pinocytosis and receptor-mediated endocytosis [35,36]. This pathway plays a significant role in maintenance of amino acid nutrition in cells because of the entry of extracellular proteins followed by proteolysis in lysosomes with the delivery of the resultant amino acids to the cytoplasm. In some ways, this is akin to autophagy, which uses cellular proteins to maintain amino acid nutrition under specific conditions, again involving lysosomal proteolysis for generation of free amino acids. Interestingly, for some unknown reasons, primary emphasis is placed on macropinocytosis in amino acid nutrition, but this endocytic process cannot be limited to the uptake of extracellular proteins because of its involvement in the non-specific uptake of all components present in extracellular fluid. There is convincing evidence indicating the participation of macropinocytosis in the uptake of native as well as oxidized low-density lipoproteins (LDL) [37,38]. Therefore, macropinocytosis is also likely to provide lipid nutrients such as cholesterol, triglycerides and phospholipids to cells. Understandably, macropinocytosis and autophagy are critical for cancer cells under conditions of nutrient deprivation [39,40,41]. The importance of these alternative modes of amino acid delivery to cancer cells is underscored by the increased demand for amino acids and other nutrients to support the rapid proliferation in these cells. This is in addition to the marked upregulation of selective transporters for amino acids, peptides and other nutrients in cancer cells to satisfy the nutritional needs by the traditional pathway, namely transporter-mediated delivery [42,43,44,45,46].

Macropinocytosis is activated by various oncogenes such the EGF receptor and activating mutations in KRAS [47,48]. Even though macropinocytosis is independent of clathrin and caveolin, the process still requires remodeling of the cytoskeletal protein actin [49]. One of the major factors that positively influences this remodeling, which is necessary for the initiation of invagination of the plasma membrane for macropinocytosis, is the alkalinization of the pH on the cytoplasmic side of the plasma membrane. The activation of macropinocytosis by the EGF receptor involves the induction of the Na^+^/H^+^ exchanger subtype NHE1 (SLC9A1) that mediates the efflux of H^+^ from cells in exchange for the influx of Na^+^ [50]. Alternative signaling mechanisms may also participate in the induction of macropinocytosis by EGF [51]. In the case of KRAS mutations, it is the recruitment of vacuolar ATPase to the plasma membrane that accomplishes H^+^ efflux from the cells [52]. In both cases, the end result is the alkalinization of the cytoplasmic domain of the plasma membrane, which then promotes actin remodeling to initiate macropinocytosis.

If efflux of H^+^ from the cells either via the Na^+^/H^+^ exchanger or v-ATPase promotes macropinocytosis, could amino acid transporters that mediate H^+^ from the cells as a part of their transport mechanism have a similar effect? This question led to the investigations of the amino acid transporter SLC38A5 regarding its potential connection to macropinocytosis [53]. SLC38A5, also called System N2 (SN2) or sodium-coupled neutral amino acid transporter 5 (SNAT5), is a Na^+^-coupled transporter for the amino acids glutamine, histidine, asparagine, glycine, serine and methionine, and its transport process is coupled to a simultaneous release of H^+^ from the cells (Figure 1). The Na^+^:H^+^ stoichiometry is 1:1, which makes the transport process electroneutral because all of its amino acid substrates are zwitterionic with no net charge. As such, SLC38A5 is an amino acid-dependent Na^+^/H^+^ exchanger [54,55]. Cellular uptake of amino acids via this transporter in the presence of extracellular Na^+^ does lead to intracellular alkalinization resulting from H^+^ efflux [55]. This transporter is highly upregulated in breast cancer, particularly in triple-negative breast cancer [56]. Since the substrate selectivity of SLC38A5 includes glycine, serine and methionine, the amino acids essential for one-carbon metabolism, and also glutamine, the amino acid obligatory for the cancer cell-specific metabolic pathway known as glutaminolysis, the increased expression and activity of this transporter in cancer cells fuels oncogenic metabolism and supports tumor growth [56]. Since triple-negative breast cancer cells express high levels of SLC38A5, the connection of this transporter to macropinocytosis was investigated in these cells [53]. These studies established convincingly that amino acid uptake into cells via this transporter is coupled to activation of macropinocytosis as monitored by the cellular uptake of TMR (tetramethylrhodamine)-dextran, a fluorescent marker which detects macropinocytosis. Interestingly, SLC38A5-stimulated macropinocytosis is inhibitable by the amiloride derivatives such as ethylisopropylamiloride [53]. This raised the question as to whether this amino acid transporter, which functions as a Na^+^/H^+^ exchanger in the presence of amino acids, is inhibitable by EIPA and other amiloride derivatives which are known for their activity as inhibitors of the classical Na^+^/H^+^ exchangers. Subsequent experiments showed that it is indeed the case; SLC38A5 is directly inhibited by amilorides [53].

Until recently, four amino acid transporters received most of the attention for their role in amino acid nutrition in cancer cells; these are SLC7A5, SLC1A5, SLC7A11 and SLC6A14 [42,43,44,45]. None of these has been shown to be associated with macropinocytosis. This makes SLC38A5 unique. This transporter is also upregulated in specific cancers, and its tumor-promoting functions are not restricted to the supply of selective amino acids to cancer cells. Its function is also coupled to regulation of intracellular pH because of the involvement of H^+^ as one of the cotransported ions. Influx of amino acid substrates via SLC38A5 is associated with removal of H^+^ from the cells, thus providing a novel mechanism for the maintenance of cellular pH. This process is critical for cancer cells because they generate large amounts of lactic acid via aerobic glycolysis and hence need effective pathways to remove H^+^ from cells [57]. SLC38A5 provides one such mechanism. The new findings that this amino acid transporter also promotes macropinocytosis underscores the importance of this transporter to tumor growth because macropinocytosis is an efficient pathway for the provision of amino acid and other nutrients to cancer cells. Analysis of data available at the Cancer Genome Atlas (TGCA) reveals that SLC38A5 is upregulated not only in triple-negative breast cancer but also in pancreatic cancer. In fact, the levels of SLC38A5 expression correlate reciprocally with the survival of the patients with pancreatic cancer. It is important to note that macropinocytosis has received the utmost attention in pancreatic cancer because of the widespread occurrence of activating mutations in KRAS and their role in the potentiation of macropinocytosis as a novel mechanism to ensure optimal nutrition in cancer cells. Therefore, SLC38A5, as not only the provider of amino acids but also as an activator of macropinocytosis, assumes a unique place among the amino acid transporters that are known to be upregulated in cancer.

SLC38A3, also known as SN1 or SNAT3, represents another subtype of the amino acid transport system N [14]. The mechanism of transport function is identical for SLC38A5 and SLC38A3 in that the latter is also an amino acid-dependent Na^+^/H^+^ exchanger. We predict that SLC38A3 is also capable of activating macropinocytosis in an amino acid-coupled manner. SLC38A3 plays an obligatory role in the kidney in the acid–base balance. During metabolic acidosis, the kidney has to eliminate the excess H^+^, and this requires a source of ammonia, which combines with H^+^, and the resultant NH_4_^+^ is eliminated across the apical membrane into urine. Glutamine constitutes this ammonia source, and SLC38A3 is the primary provider of this glutamine. Accordingly, SLC38A3 is induced in the basolateral membrane of the epithelial cells in the kidney during metabolic acidosis to provide this glutamine via Na^+^-coupled uptake from the circulation [58]. Interestingly, this increased expression is specific to SLC38A3; the expression of SLC38A5 in the kidney is not influenced by metabolic acidosis [58]. A loss-of-function mutation in Slc38a3 causing Slc38a3 deficiency in a mouse model confirmed the critical role of this transporter in metabolic acidosis [59]. We speculate that macropinocytosis might be activated in kidney tubular cells during metabolic acidosis and that cellular uptake of plasma proteins and lipoproteins might increase as a result. What this means in terms of tubular cell physiology and pathology needs to be investigated. At present, this line of thinking is only speculative, and needs experimental testing and validation.

SLC38A3 and SLC38A5 are also expressed in the brain, primarily in astrocytes where they are considered to play a role in the glutamine–glutamate cycle in which the transporters mediate the release of glutamine from astrocytes [60,61]. SLC38A5 is also expressed specifically at markedly high levels in endothelial cells of the blood–brain barrier and blood–retinal barrier [62] (https://www.proteinatlas.org/ENSG00000017483-SLC38A5/single+cell+type/eye, accessed on 1 January 2022). It would be interesting to ascertain in future studies if these transporters promote macropinocytosis in these cell types and if they do, what the physiological and pathological consequences are from the resultant entry of extracellular components into these cells.

## 4. Obesity and SLC6A19: Deficiency of SLC6A19 Protects against Diet-Induced Obesity/Metabolic Syndrome in Mice via Increased Secretion of FGF21 and GLP-1

SLC6A19 is a Na^+^-coupled transporter for neutral amino acids that plays a major role in the intestinal absorption of dietary protein-derived amino acids and in the renal reabsorption of circulating amino acids filtered at the glomerulus [63]. Trafficking of this transporter to the luminal membrane of the epithelial cells of the small intestine and proximal tubule requires a chaperone, and this function is fulfilled by ACE2 in the intestine and collectrin in the kidney [31]. Loss-of-function mutations in SLC6A19, the “transporter proper”, cause Hartnup disease, an inborn error of amino acid transport [64]. The consequences of this defective transport function manifest in the kidney in the form of increased urinary excretion of neutral amino acids and in the intestinal tract in the form increased passage of neutral amino acids into the ileum and colon. Elimination of tryptophan, one of the neutral amino acids, in the urine may cause tryptophan deficiency, hence resulting in the deficiency of the vitamin niacin. This is because a significant portion of niacin in our body arises from endogenous synthesis from tryptophan. As a consequence, niacin deficiency (i.e., pellagra) is a major, and probably the only, clinical symptom associated with this disorder. Even this manifestation might depend on the protein content in the diet; patients with high protein intake in their diet may not show any symptoms of pellagra. In contrast, patients with suboptimal dietary intake of proteins might be at risk for not only pellagra but also broad-spectrum neurological complications because of the need for many of the SLC6A19 substrates such as tryptophan, phenylalanine and tyrosine for the synthesis of the neurotransmitters serotonin, dopamine and norepinephrine in the brain. Patients with Harnup disease are likely to have a deficiency of these amino acids due to increased elimination in urine.

An intriguing and interesting observation was made when SLC6A19-null mice were studied as a model for Hartnup disease [65,66]. The mice did show increased urinary excretion of neutral amino acids as expected; they also showed reduced post-prandial levels of certain neutral amino acids in circulation, particularly when the animals were fed a high-protein diet [65]. The more interesting observation was, however, the significantly improved glycemic control in the null mice as evident from the glucose tolerance test and insulin tolerance test [66]. This metabolic phenotype was associated with elevated levels of fibroblast growth factor FGF21 and the gut hormone glucagon-like polypeptide GLP-1. Based on the known physiological functions of these two hormones, this hormone profile is in accordance with various biological parameters observed in the null mice, including efficient disposal of blood glucose, reduced adipose tissue mass but increased brown adipose tissue and decreased glucose output in liver. These novel findings suggest that SLC6A19 could potentially be exploited as a useful therapeutic target for obesity, type 2 diabetes and metabolic syndrome [67,68]. The rationale is as follows: if the deletion of SLC6A19 protects against obesity/metabolic syndrome and improves glycemic control, a similar metabolic outcome could be achieved with pharmacological inhibition of the transporter.

It has already been shown that the loss of SLC6A19 in mice leads to increased circulating levels of FGF21 and GLP-1 [66]. FGF21 is secreted by the liver, but other organs also secrete this peptide. Even though the mechanisms underlying the increased secretion of FGF21 by the liver in SLC6A19-null mice are not known, it is believed that decreased delivery of amino acids from the small intestine to the liver via portal blood might be responsible for this effect [66]. FGF21 is known to be secreted in response to starvation and other stresses [69,70], and it is likely that the decreased delivery of amino acids to the liver is recognized as starvation by liver cells. The well-established biological functions of FGF21 as a key regulator of energy balance, glucose homeostasis, fat metabolism and insulin sensitivity provide clues as to the SLC6A19 loss-associated protection against obesity and metabolic syndrome. These effects of FGF21 are mediated by the activation of the heterodimeric receptor complex consisting of the FGF21 receptor and β-klotho. The involvement of the increased levels of FGF21 in protecting against obesity/metabolic syndrome observed in SLC6A19-null mice is supported by numerous recent studies highlighting the therapeutic potential of FGF21 receptor agonists for the treatment of obesity-related diseases such as type 2 diabetes and non-alcoholic fatty liver [69,70].

When dietary protein-derived amino acids are not absorbed efficiently in the proximal small intestine, they end up in the ileum and colon. These amino acids, when present in excess, might function as signals for the secretion of the gut hormones such as GLP-1 by the enteroendocrine cells located in the terminal portion of the small intestine and in the large intestine (Figure 2). There are several subtypes of enteroendocrine cells distributed differentially along the entire length of the intestinal tract, and these different subtypes secrete different gut hormones. Examples include G cells secreting gastrin in the antral region of stomach, S cells secreting secretin in duodenum, I cells secreting cholecystokinin also in duodenum and L cells secreting GLP-1 in the ileum and colon. Therefore, the amino acids that escape absorption in the small intestine due to a lack of SLC6A19 reach the ileum and colon at higher-than-normal levels and consequently act on L cells to promote GLP-1 secretion. Little is known about the underlying mechanisms for amino acid-induced GLP-1 secretion, but the involvement of specific amino acid transporters is likely. One likely candidate is SLC6A14. The expression pattern for this transporter in the intestinal tract is unique; it is expressed at relatively lower levels in small intestine, and its expression increases towards the ileum and colon [71]. The transporter protein appears to be present uniformly in the entire epithelial cell layer of the colon. Therefore, even though its expression has not yet been documented in L cells, it is likely to be present in these cells to signal excess food intake and to promote the secretion of satiety hormones such as GLP-1. Such a mechanism could explain the increased circulating levels of GLP-1 in mice in response to the deletion of SLC6A19.

The potential crosstalk between these two amino acid transporters in controlling the secretion of the satiety-inducing hormone GLP-1 is of physiologic importance. The differential expression of SLC6A19 in the proximal part of the intestinal tract in contrast to the expression of SLC6A14 in the distal part is critical for this crosstalk. The effectiveness of this functional collaboration between the two transporters to control food intake and insulin secretion might also depend on protein content in the diet. When the dietary protein content is within normal limits, SLC6A19 may be sufficient to mediate efficient absorption of the protein-derived amino acids, thus preventing their entry into the ileum and colon to signal through SLC6A14. In contrast, when the dietary protein intake is high, the resultant amino acids in the lumen of the small intestine might overwhelm the transport capacity of SLC6A19, thus letting a significant number of amino acids overflow into the distal portion of the small intestine and into the large intestine despite the normal activity of SLC6A19, thus eliciting signals via SLC6A14 to promote hormone secretion from L cells.

SLC6A14 may not be the only mechanism for amino acid signaling in L cells to control GLP-1 secretion. Cell-surface receptors might also play a role. Tryptophan and phenylalanine serve as agonists for the G-protein-coupled receptor GPR142, which is expressed throughout the intestinal tract including the colon, and promote GLP-1 secretion [72,73]. Therefore, the increased GLP-1 secretion observed in SLC6A19-null mice could arise not only from SLC6A14-mediated depolarization associated with amino acid entry in L cells but also from signaling via GPR142 without amino acid entry into cells. With SLC6A14, cellular depolarization associated with amino acid transport might open up voltage-gated calcium channels to increase intracellular Ca^2+^ levels, thus promoting exocytosis of hormone-loaded vesicles. In the case of GPR142, the most likely intracellular signaling responsible for the initiation of hormone secretion is the G_o/q_-mediated increase in Ca^2+^ levels [72,73]. In the context of the possible involvement of GPR142 in protecting against obesity/metabolic syndrome seen in SLC6A19-null mice, it is important to highlight that tryptophan and phenylalanine, the amino acid agonists for the receptor, are among the preferred substrates for SLC6A19. It is also necessary to emphasize here that the participation of SLC6A14 and GPR142 is not mutually exclusive; both pathways could operate simultaneously to regulate hormone secretion from L cells by luminal amino acids in the colon.

## 5. Deficiency of SLC6A14 Promotes Diet-Induced Obesity and Metabolic Syndrome in Humans and Mice: SLC6A14 Differs from SLC6A19 with Regard to Obesity

Amino acids in the gut lumen and in blood constitute an important determinant in the secretion of the appetite-regulating hormones. A rise in luminal or plasma levels of amino acids following food intake could be sensed by an amino acid transporter whose mediation of amino acid transport into endocrine cells in the gut, pancreas and brain could provide signals for the release of appetite-regulating hormones. Recent studies have implicated the transporter SLC6A14 in this process [74,75,76,77]. One specific single-nucleotide polymorphism (SNP) in this gene has been associated with obesity in multiple cohorts (Swedish, French and Finnish). SLC6A14 is expressed in the gut (ileum/colon) where L-type enteroendocrine cells that secrete GLP-1, peptide YY and pancreatic polypeptide are present, and in specific cell types in pancreatic islets that secrete pancreatic polypeptide (https://www.proteinatlas.org/ENSG00000268104-SLC6A14/single+cell+type/pancreas, accessed on 1 January 2022). Interestingly, this transporter is expressed in the pituitary but not in the hypothalamus, where appetite-controlling hormones are secreted in the brain [78]. These findings pinpoint the gut and pancreas, not the brain, as the major sites of action for SLC6A14 in the regulation of appetite. We hypothesize that SLC6A14 in L-type enteroendocrine cells in the ileum/colon and in PP cells in the pancreas senses amino acid levels in gut lumen and plasma, respectively, cause transport-associated membrane depolarization in these cells and promote secretion of anorexigenic hormones. We have shown that the obesity-associated SNP, present in the 3′-UTR of SLC6A14 mRNA, decreases the expression of the transporter protein [79]. A decrease in SLC6A14 transport activity would break the link between the amino acids and the secretion of appetite-suppressing hormones. As such, the appetite–satiety switch becomes malfunctional, and anorexigenic hormones are not secreted following a meal, thus causing continued food intake and weight gain. 

We have validated the cause–effect relationship between SLC6A14 deficiency and increased food intake/obesity with evidence that SLC6A14-null mice are at increased risk for diet-induced obesity, fatty liver and metabolic syndrome [79]. First, mice lacking SLC6A14 appear phenotypically normal when fed a normal rodent diet and reproduce normally. In terms of plasma amino acids, there were only subtle differences between wildtype and null mice. The only amino acids which show a significant decrease in plasma are glutamine and glycine, both of which are very good substrates for the transporter. Based on the preferential expression of this transporter in the colon, it is tempting to speculate that the transporter might actually participate in the colonic absorption of bacteria-derived amino acids, thus explaining the differences in circulating levels of at least some of the amino acids. The difference in body weight becomes apparent only when fed a high-fat diet; the SLC6A14-null mice gained more weight than wildtype mice. The age-matched null mice did weigh slightlymore than the wildtype mice even when fed the normal diet, but the magnitude of the difference was small. The obesity associated with SLC6A14 loss on a high-fat diet is accompanied with many parameters indicative of metabolic syndrome. This includes elevated fasting plasma glucose, altered glucose and insulin tolerance tests indicative of insulin resistance and increased circulating levels of insulin and leptin. The null mice also exhibited increased food intake but no change in locomotor activity and respiratory exchange ratio. Obesity is often associated with a wide spectrum of liver abnormalities; this is also true with obesity, as seen in SLC6A14-null mice on a high-fat diet. The changes in liver, namely excessive accumulation of fat, increased liver weight, increased signs of inflammation and fibrosis observed in the null mice are all characteristic of non-alcoholic fatty liver disease. In addition, there was increased epididymal fat in male mice and abdominal fat in female mice. The increased deposition of fat in the liver in null mice correlates with appropriate changes in the expression of genes involved in fatty acid uptake and fat metabolism.

The molecular mechanisms underlying the obesity in SLC6A14-null mice are not known. We speculate that the transporter is expressed in L cells present in the ileum/colon and that these cells sense the presence of amino acids in the ileal and colonic lumen via this transporter that mediates electrogenic entry of these amino acids into cells accompanied with membrane depolarization, Ca^2+^ entry and exocytosis of hormones into blood (Figure 2). This is only speculative at present; further research is needed to prove or disprove this idea. It is important to note that the transport function of SLC6A14 has also been shown to be relevant as a determinant of clinical phenotype in other diseases [80].

## 6. Significance of SLC6A19 and SLC6A14 to Obesity in Humans

The association of SLC6A19 deficiency to protection against obesity has thus far been demonstrated only in mice. Whether such association exists in humans is not known. Loss-of-function mutations, which cause Hartnup disease, have been identified in humans, but the issues of food intake, body weight, insulin resistance and glucose homeostasis have not been evaluated in these patients. Therefore, it is not known at present whether SLC6A19 deficiency would protect against obesity and obesity-associated clinical sequelae in humans. The scenario is different for SLC6A14. A specific genetic polymorphism in the SLC6A14 gene, though present in a non-coding region, suppresses the expression of the gene, and this polymorphism correlates with obesity. Deletion of SLC6A14 in mice confirmed that SLC6A14 loss leads to diet-induced obesity/metabolic syndrome and fatty liver. These findings strongly indicate the clinical significance of the association between obesity and SLC6A14 to humans.

## 7. Taurine and Its Transporter in Connection with Obesity

Taurine is not a proteinogenic amino acid, and it contains a sulfonic acid group (SO_3_H) instead of a carboxylic acid group (COOH). Furthermore, unlike the proteinogenic amino acids, which are all α-amino acids, taurine is a β-amino acid, meaning that the amino group is attached to the β-carbon atom. Interestingly, notwithstanding the fact that it is not found in proteins, it represents the most abundant amino acid in plasma and in certain specific tissues such the heart, brain, retina and placenta. Intracellular concentrations of taurine in these tissues could be as high as 10–15 mM, thus making a significant contribution to cellular osmolality. As such, this amino acid is noted as an osmolyte, and its intracellular concentrations are modulated in response to extracellular osmolality. There is also a widespread belief that taurine promotes contractile function of skeletal and cardiac muscles, thus encouraging its commercial use as a nutritional additive in many energy drinks. The biological importance of taurine is evident from studies on cats, which are not capable of endogenous synthesis of this amino acid and hence must solely depend on dietary sources. When cats are fed a taurine-free diet, they become taurine-deficient with life-threatening consequences, ultimately resulting in death if the intake of the taurine-deficient diet continues. Without taurine in the diet, cats become blind and suffer from severe cardiomyopathy [81,82]. Taurine deficiency in humans and other animals is rare because of their ability to synthesize this amino acid endogenously, thus minimizing the nutritional relevance of dietary sources. There is a specific transporter for this amino acid, and this taurine transporter (SLC6A6) is highly energetic with its transport process coupled to three different driving forces, namely a Na^+^ gradient, a Cl^−^ gradient and membrane potential [83,84]. The multiple driving forces coupled with the fact that taurine is not metabolizable explain the accumulation of this amino acid inside the cells at millimolar concentrations despite the fact that the extracellular concentration in plasma is only around 50 μM. The same transporter is also responsible for the effective absorption of dietary taurine in the small intestine and for efficient reabsorption in the kidney following filtration of circulating taurine at the glomerulus. Despite the findings that taurine deficiency does severely compromise the functions of the retina and cardiac muscle, the potential biological functions of this amino acid are known only in general terms such as osmoregulation, neuromodulation, membrane stabilization, calcium flux regulation, bile acid conjugation and protection against oxidative stress [81,82]. Intriguingly, very little is known about the biological functions of taurine at xmolecular level.

Unfortunately, this also applies to the role of taurine and its transporter SLC6A6 in obesity. There is increasing evidence for an association of plasma levels of taurine and the expression levels of its transporter with obesity and body mass index, indicating some role of this amino acid in body weight control, adipocyte biology, insulin secretion and glucose homeostasis [85]. However, specifics of this association are lacking at the molecular level. Animal and human studies have shown that taurine might serve as an anti-obesity agent, negatively affecting the gain of body mass and fat mass, accumulation of fat in liver, plasma lipids, inflammation and insulin resistance. A detailed study of nutrient transporter expression in human jejunum has revealed a strong negative correlation between the intestinal expression of the taurine transporter SLC6A6 and body mass index [86]. Placenta shows robust activity of SLC6A6 to transfer maternal taurine to the developing fetus [87,88], and maternal obesity is associated with a reduction in placental taurine transporter activity [89]. Furthermore, SLC6A6-null mice are lean and show resistance to diet-induced obesity [90,91]. Recently, the presence of mutations in SLC6A6 in association with retinal degeneration has been described in humans [92], but there is no evidence of extraocular manifestations in these patients. Additional research is needed to gain a better understanding of the association between the taurine–taurine transporter and obesity.

## 8. Conclusions

It is becoming increasingly apparent that the biological functions of amino acid transporters in mammalian cells go far beyond their expected role in transmembrane transfer of amino acids. They seem to be involved in multiple non-traditional biological processes such as virus entry, intracellular signaling, modification of clinical phenotype in diverse diseases, obesity, diabetes, fatty liver and metabolic syndrome. The recent discovery of the role of SLC38A5 in promoting macropinocytosis and the contrasting roles of SLC6A19 and SLC6A14 in diet-induced obesity and metabolic syndrome highlights these novel and unconventional functions of specific amino acid transporters. Based on these newly recognized functions, it seems reasonable to expect that some of the amino acid transporters have potential for exploitation in the future as drug targets for the treatment of selective diseases.

## Figures and Tables

**Figure 1 biomolecules-12-00235-f001:**
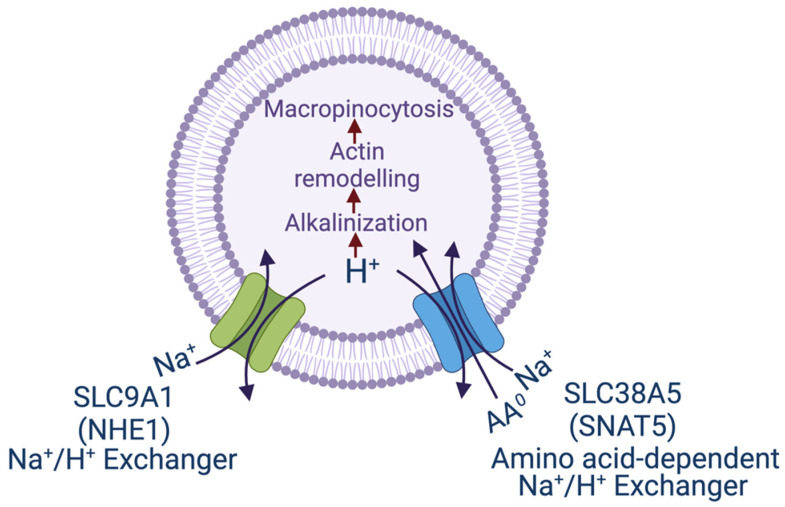
Transport pathways for NHE1 and SNAT5 and their relevance to macropinocytosis. The scheme shows the similarity between the two transporters in the efflux of H^+^, one being just the Na^+^/H^+^ exchanger (NHE1) and the other being an amino acid-dependent Na^+^/H^+^ exchanger. In both cases, the transport process leads to alkalinization in the cytoplasmic subdomain underneath the plasma membrane, which initiates remodeling of actin filaments and consequently promotes macropinocytosis. SNAT5, sodium-coupled neutral amino acid transporter 5, another name for SLC38A5.

**Figure 2 biomolecules-12-00235-f002:**
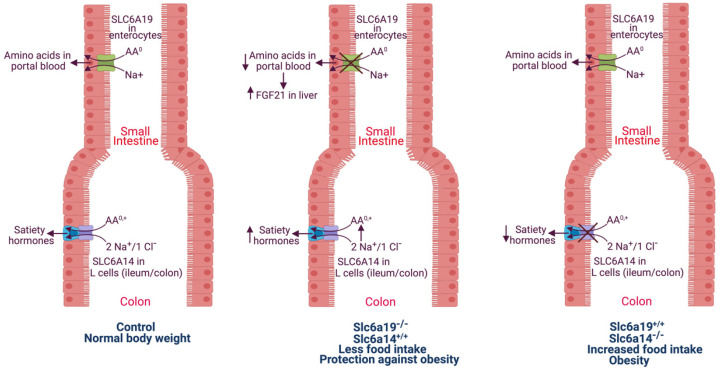
Functional crosstalk between SLC6A19 in the proximal small intestine and SLC6A14 in ileum and colon in the control of food intake and body weight. AA^0^, neutral amino acids; AA^0,+^, neutral and cationic amino acids; FGF21, fibroblast growth factor 21.

**Table 1 biomolecules-12-00235-t001:** Amino acid transporters moonlighting as cell surface receptors for cellular entry of specific viruses.

Transporter	Species	Virus
Cat 1 (Slc7a1)	Mouse	Ecotropic murine leukemia virus; Bovine leukemia virus
ASCT 1 (SLC1A4)	Human	Feline endogenous retrovirus RD-114; Baboon endogenous retrovirus
ASCT 2 (SLC1A5)	Human	Baboon endogenous retrovirus; Human endogenous retrovirus HERV-W
ACE 2 (chaperone for intestinal amino acid transporters SLC6A19 and SLC6A20)	Human	COVID-19 virus SARS-CoV-2

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
