# Peer review of "Unconventional Functions of Amino Acid Transporters: Role in Macropinocytosis (SLC38A5/SLC38A3) and Diet-Induced Obesity/Metabolic Syndrome (SLC6A19/SLC6A14/SLC6A6)"

_biomolecules, 2022, doi:10.3390/biom12020235_

Round 1

Reviewer 1 Report

This is a review article on selected mammalian amino acid transporters and their unconventional (beyond amino acid transport) functions. The Authors list the amino acid transporters involved in the process of endocytosis of viruses (Transceptors). They further focus on the involvement of selected transporters in the process of micropinocytosis, demonstrating in an elegant way promotion of this process by alkalinization of the cytoplasmic domain of the plasma membrane. They propose that SLC38A5 and SLC38A3 can be the amino acid-dependent Na+/H+  exchangers. The second part of this review is devoted to the obesity syndrome. Apart from the quoted experimental data, the Authors present a model of a functional crosstalk between SLC6A19 and SLC6A14 regulating the SLC6A14-dependent release of fibroblast growth factor 21. They as well discuss the role of SLC6A14 and GPR142 receptor in secretion of a gut hormone GLP-1.

In the last part they sum-up the known facts on correlation between function of a taurine transporter (SLC6A6) and obesity and they propose additional research on this protein for a better understanding its role in obesity.

This is a very well written interesting review with the proposed models showing amino acid transporters as potential targets for treatment of certain pathological states.

Author Response

We thank you very much for the positive comments and for liking the review as it was in the original format.

Reviewer 2 Report

Dear Authors,

This article, “Amino acid transporters as regulators of micropinocytosis (SLC38A5/SLC38A3) and diet-induced obesity/metabolic syndrome (SLC6A19/SLC6A14/SLC6A6)” by Bhutia et al., includes an important information about the relationship of solute carrier (SLC) transporters with cellular endocytosis and metabolic diseases. For the readers, several points should be improved.

  1. Introduction: in this section, only 2 references have been cited. Certainly, this section is mainly constructed by the general information. However, the authors should include proper reference/book citations in the section as many as possible for the readers.

  1. Section 2: the descriptions seem to be important for the readers. However, it seem to be also difficult for the readers to understand these contents easily. It is recommended that the schematic diagram with summarized table is added.

  1. Section 3: the major content is expected as “macropinocytosis”. On the other hand, Figure 1 in the section only included transport manner of NHE1 and SNAT5. In addition, the information of other transporters is also included in the section. If the author prepare the figure, the summarized information should be shown as the Figure.

  1. Section 4: titles of sections 4 and 5 are almost the same. In addition, subsection in the Section 4 is only settled as #4.1. It is recommended that the description of subsection 4.1 should be deleted and the title of section 4 should be modified with the information of subsection 4.1 title.

  1. Sections 4 and 5: it is expected that the comparison of obtained evidence about SLC6A19 and 6A14 is helpful for the readers to understand their importance. The summary of the previous evidence should be added as a table, or prepared as a schematic diagram.

  1. Conclusion: it seems that the content are too general and are not focused on the transporters in the manuscript such as several SLC38As and SLC6As. It is considered that these transporters focused on the manuscript should be appealed in the conclusion section.

Thank you,

Author Response

  1. As suggested by this reviewer, we have now added nine new references in the Introduction section.
  2. We agree with this comment. We have now added a Table to summarize the involvement of specific amino acid transporters as cell-surface receptors for certain retroviruses.
  3. We agree with this comment also. We have now modified Figure 1 to indicate how the two transporters promote macropinocytosis.
  4. We have now removed the sub-section 4.1. We also changed the caption for section 4.
  5. We have also changed the caption for section 5 to highlight the contrast between SLC6A19 and SLC6A14 in their roles in obesity.
  6. We have modified the Conclusion to highlight the amino acid transporters that form the central theme of this review.

Reviewer 3 Report

Amino acid transporters are proteins located on the cell membrane, which are involved in the transport of important amino acids inside and outside the cell, and play an important role in many biological events.  This paper focuses on the role of several amino acid transporters (SLC38A5/SLC38A3) and (SLC6A19/SLC6A14/SLC6A6) in macropinocytosis and dime-induced obesity/metabolic syndrome.  
1. In the review, the role of SLC38A5/SLC38A3 in macropinocytosis was suddenly introduced, which was somewhat abrupt in logic.  
2. Macropinocytosis and diet-induced obesity/metabolic syndrome are two different events. In this paper, Why do the authors focus on amino acid transporters involved in these two events?  And there seems to be no functional crossover between the two transporters.  
3. The relationship between different amino acid transporters and these two types of events in different studies was presented in the form of tables, providing readers with more intuitive information.  

Author Response

All three comments from this reviewer appear to be related to the logic behind the focus of the current review on the role of amino acid transporters in macropinocytosis and obesity. This manuscript is submitted for a Special Issue: Recent Advances in Amino Acid Transporters. We decided to focus in our review on what we think as “unconventional” functions of certain amino acid transporters which have been identified in recent years. Yes, macropinocytosis and regulation of body weight represent distinct topics with no apparent connection, but these are the two areas where some amino acid transporters have been shown to play a role in an “unconventional” way. To highlight this point, we have now changed the title.

Round 2

Reviewer 2 Report

Dear authors,

I have satisfied the replies from the authors and revised manuscript.

Reviewer 3 Report

The author responded well to the reviewers' questions. Thanks!